# Microbial Community and Abundance of Selected Antimicrobial Resistance Genes in Poultry Litter from Conventional and Antibiotic-Free Farms

**DOI:** 10.3390/antibiotics12091461

**Published:** 2023-09-19

**Authors:** Camilla Smoglica, Muhammad Farooq, Fausto Ruffini, Fulvio Marsilio, Cristina Esmeralda Di Francesco

**Affiliations:** 1Department of Veterinary Medicine, University of Teramo, Loc. Piano D’Accio, 64100 Teramo, Italy; mfarooq@unite.it (M.F.); fmarsilio@unite.it (F.M.); cedifrancesco@unite.it (C.E.D.F.); 2Gesco Consorzio Cooperativo a r.l., 64020 Teramo, Italy; fausto.ruffini@amadori.it

**Keywords:** poultry, microbial community, 16S rRNA, *firmicutes*, antibiotic resistance, real-time PCR, colistin, tetracycline resistance, aminoglycosides, farming

## Abstract

In this study, a culture-independent approach was applied to compare the microbiome composition and the abundance of the antimicrobial resistance genes (ARGs) *aad*A2 for aminoglycosides, *tet*(A), *tet*(B), *tet*(K), and *tet*(M) for tetracyclines, and *mcr*-1 for colistin in broiler litter samples collected from conventional and antibiotic-free flocks located in Central Italy. A total of 13 flocks and 26 litter samples, collected at the beginning and at the end of each rearing cycle, were submitted to 16s rRNA sequence analysis and quantitative PCR for targeted ARGs. Firmicutes resulted in the dominant *phylum* in both groups of flocks, and within it, the Clostridia and Bacilli classes showed a similar distribution. Conversely, in antibiotic-free flocks, a higher frequency of Actinobacteria class and *Clostridiaceae*, *Lactobacillaceae*, *Corynebacteriaceae* families were reported, while in the conventional group, routinely treated with antibiotics for therapeutic purposes, the Bacteroidia class and the *Enterobacteriaceae* and *Bacillaceae* families were predominant. All investigated samples were found to be positive for at least one ARG, with the mean values of *aadA2* and *tet*(A) the highest in conventional flocks by a significant margin. The results suggest that antibiotic use can influence the frequency of resistance determinants and the microbial community in poultry flocks, even though other environmental factors should also be investigated more deeply in order to identify additional drivers of antimicrobial resistance.

## 1. Introduction

Antibiotics have been used in human and veterinary medicine for almost 70 years, and they have been shown to be effective for treatment of dangerous microbes and safeguarding human and animal health [1]. In the livestock industry, characterized by a high density of animals and environmental parameters favoring the microbial proliferation and other stress-inducing management factors, the use of antibiotics is necessary in order to preserve the high levels of productivity [2]. Therefore, the emergence of antimicrobial-resistant bacteria has become a major health concern in recent years, and the antimicrobial resistance (AMR) is actually considered a worldwide public health issue that involves human, animal, and environment health alongside important economic implications [3]. In food producing environments, the resistant bacteria can be positively selected by the continuous or massive use of antibiotics, and the antimicrobial resistance genes (ARGs) are often located on mobile elements and easily transferable to other microorganisms, including severe pathogens of humans and animals [4]. The dissemination of resistant pathogens and/or ARGs takes place through several drivers, such as animal products, feed, water, and farming waste (litter or manure) [5]. In order to tackle the AMR emergence, international and national authorities have developed specific action plans, introducing the total ban of antibiotics as growth promoters in livestock [6], a rigorous categorization of antimicrobials aimed to preserve the efficacy of last-resort molecules [7,8], and encouraging all efforts to significantly reduce the use of antimicrobials in human and veterinary medicine [9].

In this respect, the poultry industry represents one of the most innovative food-producing sectors; it has developed technologically advanced farming models with a significant reduction of antibiotics use or are completely antibiotic-free, able to satisfy the increasing demand of consumers for more sustainable, high-quality, and safety poultry products [10,11]. Despite this, the real efficacy of antibiotic-free farming in reducing the dissemination of resistant bacteria and ARGs is not completely understood. Published data demonstrated that some differences are evident, comparing intestinal microbial composition and ARGs abundance at the farm level; however, often, this evidence does not persist with respect to the carcasses at slaughterhouses or meat retailers, with particular regard to food-borne pathogens [12,13,14,15,16]. Therefore, continuous monitoring of the AMR profiles in poultry farming should be encouraged in order to highlight any changes or to identify additional sources of resistant microorganism dangerous for humans and animals. 

In this respect, poultry litter can be used to indirectly investigate the health status of animals and to monitor the emergence of pathogens harboring ARGs potentially transmissible to humans in the meat production chain. In addition, chicken litter is frequently used as organic fertilizer to enrich soils and crops, representing a source of environmental dissemination of antimicrobial-resistant bacteria and the relative genes [17]. The microbial composition of poultry litter has been deeply investigated with respect to any spatial or temporal modifications that can occur during the farming cycles or composting phases [17,18,19,20,21], but only few of studies focused on the modification of fecal microbioma and resistome in relation to the use of antibiotics during the rearing of animals, comparing both conventional and antibiotic-free systems [15,22,23]. 

The aim of this study was to characterize the microbial community and the abundance of a selected panel of ARGs in litter samples collected from antibiotic-free and conventional poultry flocks in order to highlight any differences in total microbial composition and genetic resistance determinants related to specific farming systems.

## 2. Results

### 2.1. Sequencing of 16S rRNA Gene and Data Analysis

Quality analysis and trimming allowed us to obtain a total of 1,810,205 sequences and 1902 features with a mean of 38,764 sequences per sample. The rarefaction curves for the samples are available as Appendix A.

The 16S rRNA gene analysis revealed the structure of microbial communities in all litter samples, except for the flock AF4, due to the DNA quality and abundance considered unsuitable for sequencing.

Firmicutes appeared to be the dominant *phylum* in both groups of flocks, with relative abundance frequency mean of 74.9%. In detail, the relative abundance frequency mean was 76.8% in AF flocks and 73.1% in C flocks.

At class level, Clostridia and Bacilli relative abundance means were similar between AF and C flocks with 45.15% and 40.19% for Clostridia, and 31.62% and 34.46% for Bacilli, respectively. The Actinobacteria class relative frequency mean was higher in AF (11.19%) compared to the C farms (1.45%), while the Bacteroidia class was higher in C farming (6.24%) with respect to the AF flocks (0.8%). Additionally, the Oligoflexia class (40.38%) was abundant only in one AF farm (AF3), while Campylobacteria, Vampirivibrionia, and Negativicutes classes were evident only in the C4 flock (Figure 1).

Considering the family level, *Clostridiaceae*, *Lactobacillaceae*, and *Corynebacteriaceae* were described with relative abundance means of 31.6%, 10.1%, and 5.1%, respectively. In detail, these family’s abundance was higher in AF flocks (44.4%, 15.4%, and 9.64%) with than in C flocks (17.8%, 3.9%, and 0.97%). In addition, *Enterobacteriaceae* and *Bacillaceae* families reported relative abundance means of 9.84% and of 8.37%, with means values of 5.72% and 7.3% in AF farms and 14.3% and 15.9% in C farms, respectively (Figure 2).

The α-diversity (within each sample type) evaluated at operational taxonomy unit (OTU) level using Faith’s phylogenetic diversity was significantly lower (*p* < 0.05) in AF than in C flocks (*p* = 0.024343), while it appears to be comparable between T0 and T1.

Permutational multi-variable analysis of variance (PERMANOVA) showed a significantly difference (*p* < 0.05) between microbial communities of sample types (β-diversity) in AF and C flocks (*p* = 0.020998); whereas there was no significant difference between T0 and T1 sampling times (*p* = 0.840816). Additionally, the non-metric multidimensional scaling (NMDS) plot supported the finding difference of gut microbiota across AF and C farms at the OTU level based on Jaccard distance, which is used as a distance metric of β-diversity (Figure 3).

### 2.2. Quantitative PCR Analysis of ARGs

All investigated samples were found positive for at least one ARG. The *aad*A2, *tet*(A), *tet*(B), *tet*(K), *tet*(M), and *mcr*-1 genes were detected in all samples except for the antibiotic-free group, in which the flock AF7 was negative for *tet*(B) in the T0 sample and for *mcr*-1 in both T0 and T1 samples. The normalized values of ARGs, based on the 16S rRNA abundance detected in each sample, were reported in Table 1. In the antibiotic-free group, the ARGs/16S rRNA copies ranged from 6.01 × 10^−7^ to 6.45 × 10, obtained for *tet*(M) and *mcr*-1 fragments, respectively. In the conventional group, this range varied from 3.39 × 10^−6^ for *tet*(M) to 3.97 × 10^4^ for *aad*A2. Overall, the highest concentrations of ARGs appeared to be distributed in conventional flocks (Figure 4).

### 2.3. Statistical Analysis

The statistical analysis showed a significantly difference (*p* < 0.05) in the means values for ARGs *aad*A2 (*p* = 0.0067) and *tet*(A) (*p* = 0.0001) between AF and C flocks.

The Spearman correlation analysis allowed us to describe a strong positive correlation between C flocks and *tet*(A) (r = 0.7945 *p* = 0.001) and *tet*(B) (r = 0.7705 *p* = 0.001). In addition, a strong positive correlation was found between *tet*(A) and *tet*(B) genes (r = 0.8130 *p* = 0.0014).

Considering the bacterial class Campylobacteria, a strong positive correlation was reported with Negativicutes (r = 0.8705 *p* = 0.001) and Lentisphaeria (r = 0.7357 *p* = 0.003). Additionally, a strong positive correlation was identified between Lentisphaeria and Desulfovibrionia (r = 0.7917 *p* = 0001), while a strong negative correlation was described between Bacilli and Clostridia (r = −7217 *p* = 0.001). At the bacterial family level, the correlations identified were reported in Appendix A.

## 3. Discussion

The current study focused on the capacity of antibiotic treatments to influence the composition of microbiota and the abundance of ARGs in broiler litter samples comparing different poultry production systems with and without the use of antibiotics. The litter sampling applied in this study represents an alternative non-invasive method by which to investigate the intestinal microbiota of the animals and the relative antimicrobial resistance determinants, allowing us to obtain fecal specimens highly representative of caecal or ileal contents [24]. In addition, the sequence analysis of microbial 16S rRNA allowed us to describe and quantify the entire microbial profile of samples under study, including uncultivable microorganisms, while the qPCR was performed to highlight the abundance of ARGs related to commonly used antibiotics in the traditional intensive farming system. Finally, this study was carried out in field conditions, without any experimental control of the environmental variables, with the specific aim of observing the effective changes of microbial composition and ARGs abundance during a commercial production cycle.

As previously reported [25,26], the 16S rRNA analysis revealed that the Firmicutes is the most common *phylum* in the fecal microbiota of poultry, and no differences between the AF and C groups and between sampling times (T0 and T1) were observed. Conversely, considering the remaining *phyla*, Actinobacteriota was evident in AF samples while Bacteriota was reported mainly in C samples. These results are consistent with the recent data published by Greene et al. [22], reporting similar frequencies of *phyla* taxonomic groups in the gut of both antibiotic-treated and untreated broilers.

The different composition of fecal microbiota in the flocks under study was more evident considering the distribution of bacterial classes and families, as supported by diversity analyses. Except for Clostridia and Bacilli, which appeared the most abundant classes in both groups, in AF flocks a higher frequency of Actinobacteria class, and *Clostridiaceae*, *Lactobacillaceae*, *Corynebacteriaceae* families were demonstrated, while in C flocks, which were routinely treated with antibiotics for therapeutic purposes, the Bacteroidia class and the *Enterobacteriaceae* and *Bacillaceae* families were predominant. These results are partially confirmed by Greene et al. [27], who reported an increased abundance in broiler intestinal microbiota of *Bifidobacterium*, *Bacteroidetes*, *Enterobacteriaceae*, and *Lactobacillus* in doxycicline-treated vs. untreated chickens, while Videnska et al. [28] observed a decreased abundance of *Bifidobacteriales*, *Bacteroidales*, and *Clostridiales* and an increase in *Enterobacteriales* and *Lactobacillales* in the fecal samples of broilers treated with tetracycline and streptomycin. More recently, the abundance of the genus *Lactobacillus* was higher in the intestinal tracts of untreated broilers (76–77%) compared to those of treated animals (30–55%) [22]. Conversely, other studies showed that the use of different antibiotics as growing promoters, including tetracyclines, seems to be ineffective in terms of significantly influencing the density of *Lactobacillus* spp. [29], but therapeutic doses of enrofloxacin are more effective in facilitating a relative abundance of multidrug-resistant *Enterobacteriaceae* strains [17]. Interestingly, lactobacilli, non-pathogenic clostridia, and corynebacteria are recognized as microorganisms useful to improving the digestive efficiency of chicken and for their ability to extract energy from feed [30,31,32]. In this respect, the relative abundances observed in AF samples indirectly suggest the presence of bacteria beneficial to preserving the intestinal health of animals. Conversely, the *Enterobacteriaceae* family, which includes important pathogenic microorganisms, such as *Salmonella enterica* and *Escherichia coli*, frequently characterized by antimicrobial resistance profiles [12,13,33], seems to be positively influenced by antibiotic treatments in C group. Indeed, the antibiotics used in broiler farming, even if administrated, when necessary, for a short time at the beginning of the cycle (at 5–14 days of animals age), may facilitate intestinal dysbiosis and suppression of humoral immunity, with both conditions being potentially useful to allowing the proliferation of opportunistic pathogens [34,35].

However, environmental variables other than the antibiotic treatments may have influenced the differences observed in AF and C flocks. It is widely recognized that fecal microbiota composition can be altered by breed, age, sex of animals, types of diet, litter, and other menagement factors [19,36] that, in field conditions, cannot be completely ruled out from the study. In order to reduce the influence of external variables, the investigated flocks have been selected within the same poultry industry, characterized by an integrated supply chain, including the feed manufacturing process. It is reasonable to assume that all animals have been exposed to similar management conditions, except for some environmental variables (animal density, temperature, and humidity) that were specific for each farming facility. Indeed, the α-diversity results highlighted a significant difference between AF and C flocks. This suggest that the microbial composition is influenced throughout the entire cycle of rearing not only by the different antibiotics but also by several environmental factors at the farm level. As if to confirm this, the higher α-diversity observed for C samples in comparison to AF samples could be explained by the short period of antibiotic therapy applied in C flocks, which may be considered insufficient to reduce the microbial richness, as has already been suggested by previous investigations [15].

Considering the ARGs abundance, the results showed a significant influence exerted by the use of antibiotic in C flocks on the distribution of *aad*A2 and *tet*(A) genes, related to aminoglycosides and tetracyclines resistance, respectively. Based on the last Italian Medicines Agency (AIFA) report, the tetracyclines represent 14% of total antimicrobial sales in poultry industry, while aminoglycosides cover 4% [37]. These values are substantially in line with European data [38]. Despite this, several aminoglycosides and tetracyclines ARGs was widely disseminated in food-producing animals litter and manure, particularly in the poultry sector and soil resistome [39,40]. Based on previous analogous investigations, the ARGs for tetracyclines and aminoglycosides antibiotics appeared highly abundant in organic fertilizer derived from poultry litter, being included in the “persistence resistome” characterized by low temporal fluctuations (over a 60-day period of composting) [41]. At the farm level, the relative abundance of tetracycline resistance genes increased with time in the poultry litter, irrespective of antibiotic treatment administrated during the rearing cycle [17]. Similarly, a more recent work highlighted that the abundance of tetracycline genes was not influenced by the use of grown promoting doses of antimicrobials, even if the aminoglycoside resistance genes tended to be higher in the broiler supplemented with chlortetracycline [29]. Other studies confirmed that the use of therapeutic doses of antibiotics in poultry farming increases the total ARGs load in intestinal tracts [15], even if the antibiotic-free farms had a wide distribution of resistant bacteria and related ARGs, as already observed in different geographical areas [23,42,43]. Noteworthy, most ARGs, including those investigated in this study, can be located in mobile genetic elements (MGE) as plasmids, trasposons, or insertion sequences, and they are able to move in different microbial niches, conferring the antimicrobial resistance profiles to different bacterial species and allowing for the accumulation and dissemination of resistant microorganisms in the environment [44]. Therefore, other extra-intestinal environmental sources of ARGs or the persistence of resistance determinants derived from previous conventional cycles carried out in the past cannot be ruled out.

Interestingly, the *mcr*-1 gene, specific to colistin resistance, was detected in both farming types, with the most abundant sample in one AF flock. The *mcr*-1 gene has been found in *Enterobacteriaceae* from animals, food, and people all across the globe [45]. In China, the *mcr*-1 gene was discovered for the first time in 2015 in cattle and raw meat samples, and in humans [46]. Additionally, *E. coli* isolates harboring *mcr*-1 from pigs, cattle, poultry, and turkey were described in France [47] and Belgium [48], as were the isolates from broilers in Germany [49]. Colistin is considered one of the last-resort antimicrobials useful to treat infections caused by multi-resistant bacteria in humans [7]. For this reason, its use in veterinary medicine has been severely restricted to only a few specific conditions [8]. Despite this, the wide diffusion of *mcr* genes in animals and environmental sources is evident, and it suggests that the mechanisms of colistin resistance maintenance are not yet fully characterized [50].

## 4. Materials and Methods

### 4.1. Study Area and Sampling Design

A total of 26 litter samples were collected from n. 7 antibiotic-free (AF) and n. 6 conventional (C) broiler flocks located in Central Italy. In AF flocks, the use of antibiotics for disease treatments was not allowed for no less than 2 years; while in C flocks, the therapeutic treatment was administered in the presence of increased mortality, generally occurring at 7–14 days after animal birth. Based on an antibiotic susceptibility test, amoxicillin, tetracyclines, or aminoglycosides were applied up to 5 days via drinking water. The environmental sampling procedure allowed us to collect samples representative of the entire area, recovering litter pools at various points in the shed, namely, at the center and four corners [16,40]. Each flock was sampled twice, i.e., at 7 (T0) and at 35–45 days (T1) (near to slaughtering). The recovered samples were stored at 4 °C until the laboratory investigations.

### 4.2. Extraction of DNA

DNA extraction from fecal specimens was carried out using the Maxwell^®^ 11 Instrument and Maxwell kit^®^ 11 Tissue DNA Purification (Promega, Milano, Italy), following the manufacturer’s instructions. The Denovix DS-11 FX spectrophotometer and fluorometer (Wilmington, Delaware, USA) were used to determine the quality and abundance of the recovered DNA.

### 4.3. Sequencing of 16S rRNA Gene and Data Analysis

The 16S rRNA genes were amplified, targeting the V3 and V4 regions [51], and the PCR products were indexed by the Nextera XT Index kit (Illumina, San Diego, California, USA) following the 16S Metagenomic Sequencing Library Preparation Guide protocol (Illumina, USA). Libraries were sequenced on the Illumina MiSeq sequencing platform (San Diego, CA, USA) via a 2 × 300 bp paired end approach. The DADA2 package from the Quantitative Insights into Microbial Ecology 2 (QIIME2 version 2019.4) program was utilized for 16S rRNA data analysis. [52,53]. The Naive Bayes and q2-feature-classifiers plugins were used to assign taxonomy categories, and the SILVA-Naive Bayes SkLearn-trained database was used to assign taxa [54]. The raw reads have been deposited under the accession number PRJNA1012581 in the National Center for Biotechnology Information (NCBI) Archive.

Microbial community characterization and α- and β-diversity statistics were realized using the software Calypso (http://cgenome.net/calypso/) [55], analyzing the microbial community composition and the quantification within each sample group as previously described [56].

### 4.4. Quantitative PCR Analysis of ARGs

The quantitative polymerase chain reaction (qPCR) was applied to find a panel of ARGs related to the antibiotics commonly used in conventional farming or those critically important in human medicine. Specifically, the target fragments *aad*A2 for aminoglycosides, *tet*(A), *tet*(B), *tet*(K), and *tet*(M) for tetracyclines, and *mcr*-1 for colistin were selected, considering analogous investigations previously carried out in the same area of study [23]. The above-mentioned fragments were quantified with SYBR Green^®^ qPCR protocols, as previously described [57,58], using a Magnetic Induction Cycler (Mic) real-time machine (BMS, Australia). Eight-point calibration curves were generated for each qPCR using ten-fold serial dilutions of the corresponding positive control, and good correlation coefficients (0.993 > R2 > 0.999) were obtained. The 16S rRNA gene amplification was included in order to normalize the abundance of ARGs in each litter sample. The copy number of target genes was calculated based on the calibration curves and the ratio of the ARG copy number to 16S rRNA [57,58]. According to Nieto-Claudin et al. [58], a negative sample was classified as any number ≤−8, considering a baseline threshold of −7.

### 4.5. Statistical Analysis

The statistical software package STATA (version Release 17) [59] was used to assess the association between the relative abundance of microbial taxa and ARGs or animal groups, applying the Student’s *t*-test and Spearman correlation.

## 5. Conclusions

In conclusion, the findings of this research showed that antibiotic use can influence the frequency of resistance determinants and the microbial community in poultry flocks. The reduction of antimicrobials could be useful to minimizing the contamination of foods of animal origin and the risk of transmission to the consumers of resistant pathogens and their genetic determinants. Finally, the metagenomic assays applied in this study, coupled with a PCR-based measurement of ARGs and a non-invasive, highly representative sampling method, may provide a valuable alternative approach to monitoring AMR dissemination in food-producing animals.

## Figures and Tables

**Figure 1 antibiotics-12-01461-f001:**
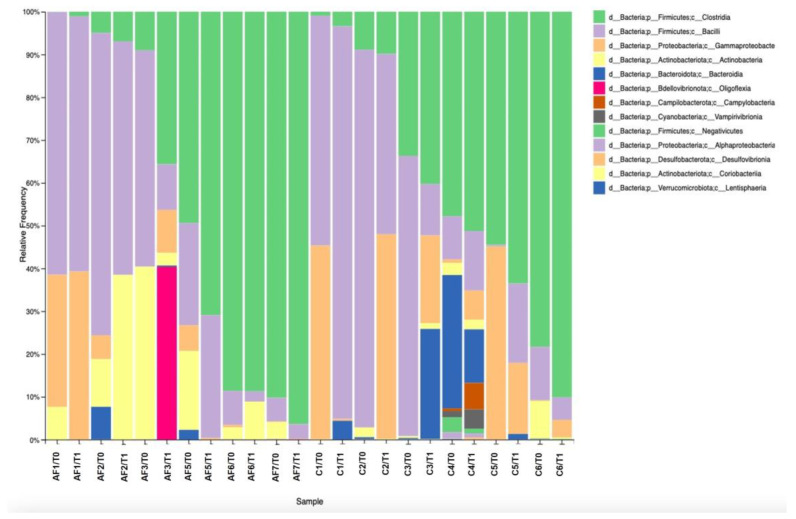
The relative frequency of microbial *phyla* classes in antibiotic-free and conventional flocks.

**Figure 2 antibiotics-12-01461-f002:**
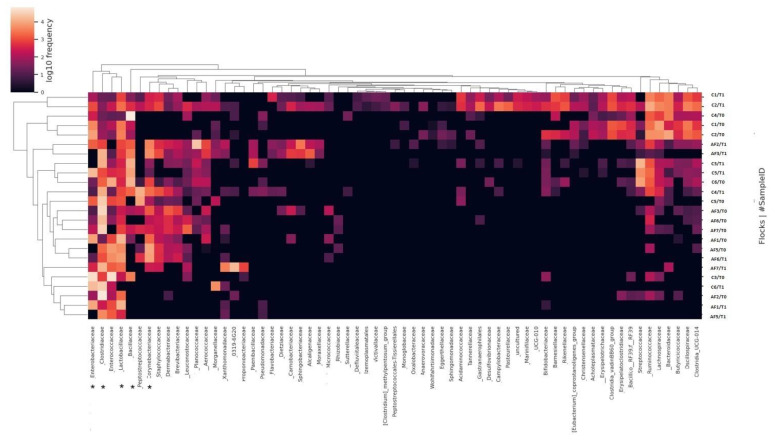
Heatmap representing the microbial community composition of antibiotic-free and conventional flocks at the family level. The asterisk shows bacterial families with different relative abundances in AF and C flocks.

**Figure 3 antibiotics-12-01461-f003:**
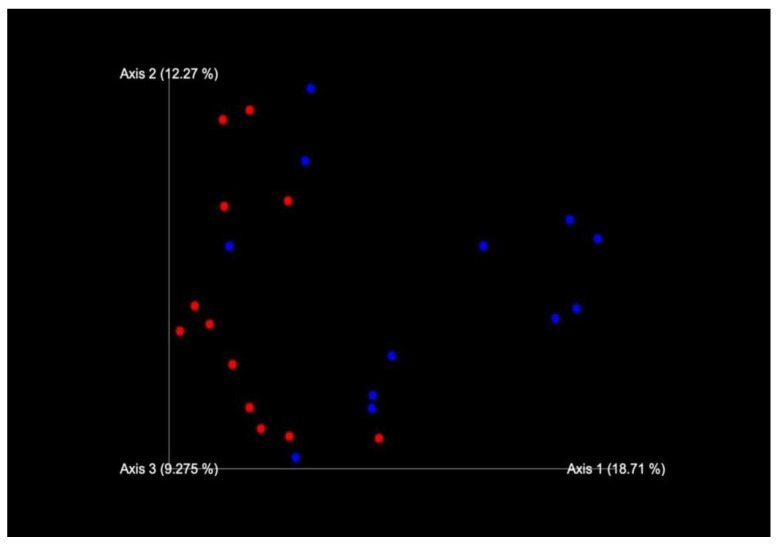
β-diversity in AF (blue dots) and C (red dots) flocks based on Jaccard distances.

**Figure 4 antibiotics-12-01461-f004:**
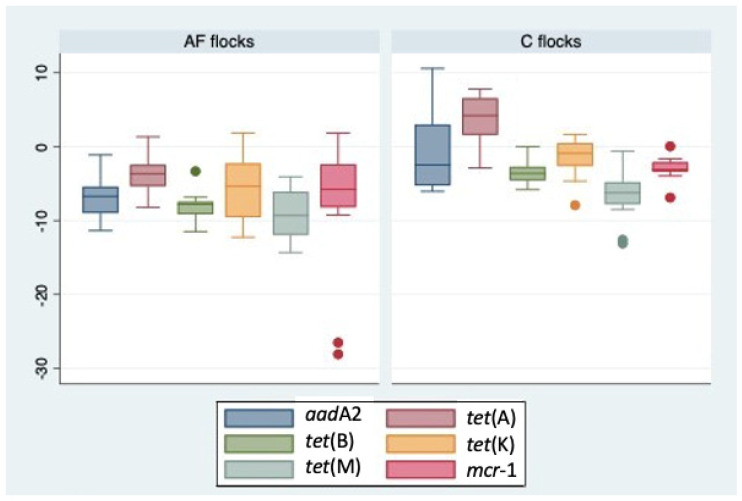
Occurrence of ARGs in AF and C flocks. Boxplots represent the 25th to 75th percentiles, and the whiskers show a maximum of 1.5× the interquartile range (IQR). The horizontal line for each boxplot represents the median value. The outliers are displayed by small dots.

**Table 1 antibiotics-12-01461-t001:** Abundance of ARGs detected in litter samples under study. The values have been normalized with respect to the number of 16S rRNA copies detected in each sample. Values ≤ 10^−8^ were considered negative.

Flock/Sample	*aad*A2	*tet*(A)	*tet*(B)	*tet*(K)	*tet*(M)	*mcr*-1
AF1/T0	4.76 × 10^−4^	4.35 × 10^−3^	1.49 × 10^−4^	1.58 × 10^−4^	2.30 × 10^−6^	9.6 × 10^−5^
AF1/T1	3.71 × 10^−3^	7.84 × 10^−2^	1.05 × 10^−4^	4.76 × 10^−6^	9.18 × 10^−5^	3.0 × 10^−1^
AF2/T0	1.32 × 10^−3^	1.84 × 10^−2^	4.92 × 10^−4^	1.16 × 10^−4^	1.88 × 10^−6^	6.5 × 10^−1^
AF2/T1	3.29 × 10^−1^	3.80 × 10	3.58 × 10^−2^	6.97 × 10^−2^	3.83 x10^−4^	6.45 × 10
AF3/T0	1.10 × 10^−3^	2.76 × 10^−4^	1.04 × 10^−5^	3.68 × 10^−5^	6.01 × 10^−7^	8.2 x10^−4^
AF3/T1	3.33 × 10^−2^	2.97 × 10^−1^	1.28 × 10^−2^	7.24 × 10^−3^	3.27 × 10^−4^	5.31 × 10^−1^
AF4/T0	2.39 × 10^−3^	3.74 × 10^−4^	1.58 × 10^−5^	1.72 × 10^−3^	2.12 × 10^−6^	9.2 × 10^−3^
AF4/T1	1.16 × 10^−5^	1.67 × 10^−2^	6.23 × 10^−5^	4.33 × 10^−5^	1.86 × 10^−5^	2.15 × 10^−3^
AF5/T0	1.11 × 10^−4^	2.41 × 10^−3^	3.35 × 10^−4^	5.64 × 10^−2^	1.27 × 10^−3^	4.45 × 10^−3^
AF5/T1	1.25 × 10^−4^	5.39 × 10^−2^	5.91 × 10^−4^	2.33 × 10^−3^	4.58 × 10^−3^	8.03 × 10^−4^
AF6/T0	2.57 × 10^−3^	9.60 × 10^−2^	4.19 × 10^−4^	1.77 × 10^−1^	8.79 × 10^−5^	1.54 × 10^−2^
AF6/T1	7.51 × 10^−2^	5.22 × 10^−3^	4.13 × 10^−4^	6.25 × 10	1.71 × 10^−2^	2.82 × 10^−2^
AF7/T0	5.37 × 10^−3^	3.61 × 10^−2^	Neg	1.54 × 10^−1^	3.95 × 10^−3^	Neg
AF7/T1	1.26 × 10^−4^	1.57 × 10^−1^	1.07 × 10^−3^	9.25 × 10^−3^	7.88 × 10^−5^	Neg
C1/T0	1.12 × 10^−1^	1.87 × 10^1^	7.20 × 10^−2^	9.51 × 10^−1^	5.56 × 10^−1^	3.18 × 10^−2^
C1/T1	1.98 × 10^3^	3.65 × 10^1^	3.09 × 10^−2^	9.49 × 10^−3^	1.65 × 10^−3^	1.01 × 10^−3^
C2/T0	2.80 × 10^−2^	1.29 × 10^1^	2.5 × 10^−2^	5.81 × 10^−2^	8.67 × 10^−4^	8.41 × 10^−2^
C2/T1	6.14 × 10^−3^	7.97 × 10^2^	2.96 × 10^−3^	8.04 × 10^−1^	1.69 × 10^−3^	3.39 × 10^−2^
C3/T0	1.92 × 10^−1^	1.87 × 10	1.18 × 10^−2^	9.04 × 10^−2^	1.25 × 10^−2^	1.91 × 10^−2^
C3/T1	2.02 × 10^−1^	1.88 × 10^3^	1.02 × 10	5.27 × 10	4.85 × 10^−2^	1.05 × 10
C4/T0	2.36 × 10^−3^	1.38 × 10^2^	9.17 × 10^−3^	3.30 × 10^−1^	2.24 × 10^−3^	3.42 × 10^−2^
C4/T1	3.36 × 10^−3^	2.39 × 10^3^	1.85 × 10^−1^	3.62 × 10^−4^	5.50 × 10^−3^	1.53 × 10^−1^
C5/T0	2.97 × 10^4^	5.59 × 10^−2^	2.88 × 10^−2^	4.97 × 10^−1^	2.01 × 10^−4^	3.91 × 10^−2^
C5/T1	3.97 × 10^4^	1.06 × 10	3.72 × 10^−3^	4.96 × 10	3.84 × 10^−3^	9.97 × 10^−2^
C6/T0	4.37 × 10^−3^	1.22 × 10^2^	5.90 × 10^−2^	2.84 × 10	2.02 × 10^−6^	5.32 × 10^−2^
C6/T1	6.51 × 10^−2^	6.24 × 10^2^	1.57 × 10^−2^	2.58 × 10^−1^	3.39 × 10^−6^	1.91 × 10^−1^

Neg.: negative samples.

## Data Availability

The data presented in this study are available on request from the corresponding author.

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
