# Peer review of "Microbial Community and Abundance of Selected Antimicrobial Resistance Genes in Poultry Litter from Conventional and Antibiotic-Free Farms"

_antibiotics, 2023, doi:10.3390/antibiotics12091461_

Round 1

Reviewer 1 Report

The manuscript raises important issues regarding the impact of the use of antimicrobial substances in chickens on the composition of the intestinal microflora and drug resistance of bacteria. The ingenuity of the authors as to the research methods used should be appreciated.

The manuscript is well written, but a few points should be clarified.

-For how many production cycles have no antibiotics/chemotherapeutic agents been used on AF farms?

- Were the same antibiotics used on each farm C? How many times and for how long were antibiotics administered during the production cycle?

-What guided the authors when selecting these (tetA/B/K/M, aadA2, mcr-1) and not other resistance genes? Why were the genes responsible for resistance to beta-lactam antibiotics (e.g. those encoding lactamases) not determined since amoxicillin was used on C farms?

Figures 1 and 2 must be corrected as they are illegible - the legend font for Figure 1 must be enlarged, as well as the font on the X and Y axes for Figure 2.

Gene names, i.e. tet(A), tet(B) etc. should be written in lower case - please correct this throughout the manuscript (text and Tab. 1).

L191 - Campylobacter does not belong to the Enterobacteriaceae family.

L120 - both samples - did you meat AFT0 and AFT1?

L117-L123, L136-139 - gene names must be written in italics

Supplementary Table 1 - set the table headers in the appropriate cells (r & p)

Author Response

Reviewer 1

Comments and Suggestions for Authors

The manuscript raises important issues regarding the impact of the use of antimicrobial substances in chickens on the composition of the intestinal microflora and drug resistance of bacteria. The ingenuity of the authors as to the research methods used should be appreciated.

We are very happy to have received a positive evaluation, and we would like to express our appreciation to Reviewer 1 for the thoughtful comments and helpful suggestions.

Our detailed, point-by-point responses to the reviewer comments are given below.

The manuscript is well written, but a few points should be clarified.

-For how many production cycles have no antibiotics/chemotherapeutic agents been used on AF farms? Were the same antibiotics used on each farm C? How many times and for how long were antibiotics administered during the production cycle?

The text has been modified on the basis of requests made by the Reviewer. (line 258-261)

-What guided the authors when selecting these (tetA/B/K/M, aadA2, mcr-1) and not other resistance genes? Why were the genes responsible for resistance to beta-lactam antibiotics (e.g. those encoding lactamases) not determined since amoxicillin was used on C farms?

As stated in the text, we have based the investigations on previous studies. We have therefore focused the research on standardised protocols for these gene targets, while also considering the potential cost implications. We very much appreciate the advice of the Reviewer and preliminary analyses are already underway to extend the study.

Figures 1 and 2 must be corrected as they are illegible - the legend font for Figure 1 must be enlarged, as well as the font on the X and Y axes for Figure 2.

The figures in the word file have been sent separately to the editorial team for final editing before publication, so they will be more legible than in this version.

Gene names, i.e. tet(A), tet(B) etc. should be written in lower case - please correct this throughout the manuscript (text and Tab. 1).

Done.

L191 - Campylobacter does not belong to the Enterobacteriaceae family.

Thank you for the correction. We apologise for the oversight and have corrected the text.

L120 - both samples - did you meat AFT0 and AFT1?

The text has been amended as requested including T0 and T1.

L117-L123, L136-139 - gene names must be written in italics

The requested changes have been made, thank you for your suggestion.

Supplementary Table 1 - set the table headers in the appropriate cells (r & p)

The requested change has been made, thank you for your suggestion.

Reviewer 2 Report

Manuscript ID: antibiotics-2622671

Title: Microbial community and abundance of selected antimicrobial resistance genes….

Authors: Camilla Smoglica. et al.

The authors describe that a culture-independent approach was applied to compare the microbiome composition and the abundance of the antimicrobial resistance genes (ARGs) aadA2 for aminoglycosides, Tet(A), Tet(B), Tet(K), Tet(M) for tetracyclines, and mcr-1 for colistin in broiler litter samples collected from conventional and antibiotic-free flocks located in Central Italy. The litter samples were submitted to the 16s rRNA sequence analysis and quantitative PCR for targeted ARGs. Firmicutes resulted the dominant phylum and Clostridia and Bacilli classes showed a similar distribution. Conversely, in antibiotic-free flocks, a higher frequency of Actinobacteria class and Clostridiaceae, Lactobacillaceae, Corynebacteriaceae families was reported while in conventional group the Bacteroidia class and the Enterobacteriaceae and Bacillaceae families were predominant. The mean values of aadA2 and Tet(A) significantly highest in conventional flocks. The results suggest that the antibiotic use can influence the frequency of resistance determinants and microbial community in poultry flocks,

Overall, this manuscript is reasonably described. However, several concerning points need to be addressed.

Comments:

First, this manuscript lacks some consideration. The author should conclude more clearly at the ending part in discussion. Especially, the reviewer can’t help but feel that there are some continuations in the discussion part.

 Second, in Fig. 1, even in antibiotic-free flocks, the intestinal flora changes between T0 and T1. Are there any effects of antibiotics on this change? And as an environmental factor, is there a possibility of some influence nearby conventional broiler farms?

In addition, the author describe that Fig. 2 explains the content of lines 99-104. However, it is difficult to understand specific numbers and trends. Therefore, the author should explain in more detail.

Minor comments:

1. The distinction between Actinobacteria and Coriobacteriia (yellow) and Gammaproteobacte and Desulfovibrionia (orange) should be made clearer in fig. 1.

2. In 6 conventional broiler flocks, the author understands how much antibiotics were used on each farm?

Author Response

Reviewer 2Manuscript ID: antibiotics-2622671

Title: Microbial community and abundance of selected antimicrobial resistance genes….

Authors: Camilla Smoglica. et al.

The authors describe that a culture-independent approach was applied to compare the microbiome composition and the abundance of the antimicrobial resistance genes (ARGs) aadA2 for aminoglycosides, Tet(A), Tet(B), Tet(K), Tet(M) for tetracyclines, and mcr-1 for colistin in broiler litter samples collected from conventional and antibiotic-free flocks located in Central Italy. The litter samples were submitted to the 16s rRNA sequence analysis and quantitative PCR for targeted ARGs. Firmicutes resulted the dominant phylum and Clostridia and Bacilli classes showed a similar distribution. Conversely, in antibiotic-free flocks, a higher frequency of Actinobacteria class and ClostridiaceaeLactobacillaceaeCorynebacteriaceae families was reported while in conventional group the Bacteroidia class and the Enterobacteriaceae and Bacillaceae families were predominant. The mean values of aadA2 and Tet(A) significantly highest in conventional flocks. The results suggest that the antibiotic use can influence the frequency of resistance determinants and microbial community in poultry flocks,

Overall, this manuscript is reasonably described. However, several concerning points need to be addressed.

 We are very happy to have received a positive evaluation, and we would like to express our appreciation to Reviewer 2 for the thoughtful comments and helpful suggestions.

Our detailed, point-by-point responses to the reviewer comments are given below.

Comments:

First, this manuscript lacks some consideration. The author should conclude more clearly at the ending part in discussion. Especially, the reviewer can’t help but feel that there are some continuations in the discussion part.

We appreciate the reviewer's suggestion, but in order not to prolong the discussion, the conclusions have been clarified in the appropriate paragraph at the end of the text, based on the journal's guidelines.

 Second, in Fig. 1, even in antibiotic-free flocks, the intestinal flora changes between T0 and T1. Are there any effects of antibiotics on this change? And as an environmental factor, is there a possibility of some influence nearby conventional broiler farms?

Although a difference would appear to be present graphically, this is not apparent from the statistical analyses carried out in both the antibiotic free and conventional farms. Therefore the authors did not consider it useful to comment on these small changes in microbial composition.

We would also like to point out that the farms are geographically separate and there is no sharing of equipment or personnel. We therefore exclude any such influence.

In addition, the author describe that Fig. 2 explains the content of lines 99-104. However, it is difficult to understand specific numbers and trends. Therefore, the author should explain in more detail.

The figure has been modified by highlighting the most represented families described in the text. 

Minor comments:

  1. The distinction between Actinobacteriaand Coriobacteriia (yellow) and Gammaproteobacte and Desulfovibrionia (orange) should be made clearer in fig. 1.

The graph editor used does not allow too many colours to be inserted, but the order of the colours in the legend respects the frequency order of the graph. Thus, the graph is read from top to bottom and the colours repeated in C4/T0 and C4/T1 at the bottom of the bars correspond to the last classes in the legend.

  1. In 6 conventional broiler flocks, the author understands how much antibiotics were used on each farm?

The text has been modified on the basis of requests made by the Reviewer 1 and 2. (lines 258-261)